# Multifaceted intervention to Reduce Antimicrobial Prescribing in Care Homes: a process evaluation of a UK-based non-randomised feasibility study

Rachel Potter,[1] Anne Campbell,[2] David R Ellard [ID] ,[1,3] Catherine Shaw,[4] Evie Gardner,[5] Ashley Agus [ID] ,[5] Dermot O'Reilly,[6] Martin Underwood,[1,3] Mark Loeb,[7] Bob Stafford,[8] Michael Tunney,[4] Carmel Hughes[4]

For numbered affiliations see end of article.

**Correspondence to**
Dr David R Ellard;
D.R.Ellard@warwick.ac.uk

## ABSTRACT

**Objectives** To explore the facilitators and obstacles to the development and implementation of the Reduce Antimicrobial Prescribing in Care Homes intervention.
**Design** We used a mixed-methods approach. We conducted focus groups with care home staff and relatives of residents, and interviews with general practitioners (GPs) and home managers, completed observational visits and collected demographic data, training attendance records and data on the use of a decision-making algorithm. We used normalisation process theory to inform topic guides and interpretation of the data.
**Setting** Six care homes, three in Northern Ireland and three in the West Midlands, England.
**Intervention** A decision-making algorithm for urinary tract, respiratory tract and skin and soft-tissue infections, plus small group interactive training for care home staff.
**Results** We ran 21 training sessions across the six homes and trained 35/42 (83%) of nurses and 101/219 (46%) of all care staff. Care home staff reported using the decision-making algorithm 81 times. Postimplementation, staff reported being more knowledgeable about antimicrobial resistance but were unsure if the intervention would change how GPs prescribed antimicrobials. The pressures of everyday work in some homes meant that engagement was challenging at times. Staff felt that some of the symptoms included in decision-making algorithm, despite being evidence based, were not easy to detect in residents with dementia or urinary incontinence. Some staff did not use the decision-making algorithm, noting that their own knowledge of the resident was more important.
**Conclusion** We delivered a training package to a substantial number of key staff in care homes. A decision-making algorithm for common infections in care homes empowered staff but was challenging to operationalise at times. A future study should consider the findings from the process evaluation to help ensure the successful implementation on a larger scale.

## INTRODUCTION
In the UK, care homes (with or without nursing) provide care for older people who can no longer live independently. The prescribing of medications is the most

### Strengths and limitations of this study

► We used a mixed-methods approach to provide an in-depth and comprehensive understanding of the implementation of a new intervention into practice.
► Normalisation process theory informed our interpretation of the data.
► The study was undertaken in six care homes, which may limit transferability of findings

frequent acute healthcare intervention received by care home residents.[1] There are concerns about the quality of prescribing for care home residents generally, and in particular antimicrobials (antibiotics, antifungals and antivirals).[1] Prescribing decisions for care home residents are, at times, made remotely by telephone, without full information, potentially leading to medicines management problems and prescribing errors.[2]

Antimicrobial prescribing in care homes is a global problem, contributing to increasing antimicrobial resistance (AMR).[1] Several reports highlight the importance of this issue and note the vulnerability to infection of older people living in care homes.[3–5] The reports emphasise the need for better stewardship of antimicrobials to minimise resistance at both patient and community levels. Education of the healthcare workforce is seen as an essential element to draw attention to AMR and antimicrobial stewardship.[3–5] It has been proposed that in care homes, nurse leaders and health professionals could act as champions for antibiotic stewardship, and support should be provided in antimicrobial prescribing decision-making to reduce excessive use.[6–8]

Interventions that promote antimicrobial stewardship should include structured assessment, communication between home staff

and prescribers, and education about AMR.[8] Loeb *et al* evaluated a multifaceted intervention to reduce antibiotic prescribing in nursing homes in Canada and the USA.[9] The intervention consisted of diagnostic treatment algorithms for urinary tract infections (UTIs), supported by small group educational interactive sessions for staff. Findings indicated that fewer courses of antimicrobials were prescribed for suspected UTIs, but no significant differences were found in total antimicrobials prescribed, hospital admissions and mortality.

In the REduce Antimicrobial prescribing in Care Homes (REACH) feasibility study, we sought to adapt this approach in the UK, and extend it to include respiratory tract infections (RTIs) and skin and soft-tissue infections (SSTIs). The development of the intervention is described in detail elsewhere.[10] Briefly, we explored evidence and expert consensus, held focus groups with staff and relatives of residents and interviews with general practitioners (GPs) to adapt the approach used by Loeb *et al*[9] to produce an intervention which composed of a decision-making algorithm (targeting the three main infections) and small group interactive training.

The decision-making algorithm started with a list of non-specific signs and symptoms of infection; this was followed by observation of the resident, to include temperature and specific signs and symptoms for each infection, and finished with instruction to care home staff on how to proceed, depending on the presenting symptoms.

The training package included information about AMR, how to use the decision-making algorithm, and how to communicate with GPs using the Situation-Background-Assessment-Recommendation (SBAR) tool.[11] Training lasted between 1.5 and 2 hours. A study handbook was also provided and for those unable to attend the training, a training presentation was available as a video on a DVD and online platform.

Following the delivery of the training package, we asked staff to use the decision-making algorithm for a period of 6 months, each time they suspected a resident had a UTI, RTI or SSTI. We implemented the intervention in six care homes, three in Northern Ireland (NI) and three in the West Midlands, England.

In this paper, we describe the process evaluation for the REACH study.

## Aim
The aim of the process evaluation was to explore the facilitators and obstacles to the development and implementation of the intervention in care homes.

## METHODS
The process evaluation was adapted from the Medical Research Council framework[12] and included four key components for process evaluations proposed by Steckler and Linnan[13]: context, reach, dose delivered and dose received. The four components help to define the extent to which an intervention is implemented and a process evaluation can also help to explain how an intervention can be optimised or why it may have failed.[12] The components of the process evaluation we used are presented in table 1.

We used a mixed-methods approach to provide an in-depth and comprehensive understanding of the implementation of the intervention into practice.

### Quantitative data
We collected data on the demographics of the homes including type of home, number of staff, number of beds, bed occupancy and age range of residents. We recorded the number of staff that attended initial and follow-up training sessions. Staff were asked to complete a 'use of decision-making algorithm' form each time they used the algorithm or when they suspected a resident may have an infection but did not use the algorithm.

### Qualitative data
#### Focus groups and interviews
Preimplementation, researchers AC and RP conducted focus groups with care home staff and relatives of residents to explore normal practice within the homes when

| Table 1 | Components of the process evaluation | |
|---|---|---|
| **Component** | **Definition** | **Data description** |
| Context | Aspects of the larger social political and economic environment that may influence implementation. | Demographics of the homes and residents. Preimplementation focus groups with care home staff and relatives of residents; interviews with GPs. Ethnographic type observations. |
| Dose delivered | The number or amount of intended units of each intervention or each component delivered or provided. | Number of training sessions delivered. |
| Reach | The proportion of the intended target audience that participates in the intervention | Number of staff trained. Postimplementation focus group with staff. |
| Dose received | The extent to which participants actively engage with and interact with the recommended resources. | Postimplementation focus groups and interviews. Use of decision-making algorithm. |

GP, general practitioner.

staff suspected a resident had an infection and to discuss the development and perceived potential of the intervention. We also conducted postimplementation focus groups with care home staff to explore their experience of the training for the study and the implementation of the intervention. For all focus groups, care home managers were asked to distribute information packs on behalf of the research team which included a participant information sheet, consent form and letter inviting them to take part in the study. The managers of the care homes were not invited to take part in the focus groups to prevent the management relationship directing or constraining the group discussion.

Preimplementation, AC and RP also conducted face-to-face semistructured interviews with GPs associated with the home to explore usual practice for the management of infections in care home residents. On agreeing to take part in the study, the manager of each care home was asked to appoint a REACH Champion, an individual who would be responsible for delivering training to staff unable to attend the original REACH training session. Postimplementation, AC and RP conducted interviews with the REACH Champions and home managers to explore their experience of the training for the study, delivering the intervention, completing the study paperwork, and facilitators and obstacles to undertaking a larger study. Participation was voluntary and written consent was obtained from participants for all focus groups and interviews.

Interview and focus group discussion guides were developed based on the four key constructs from normalisation process theory.[14 15] This sociological theory aims to explain the social processes that can lead to the routine embedding, or normalisation, of a new health organisational practice, focusing on the work that individuals and groups do to enable an intervention to become normalised. The four key constructs are: making sense (coherence), engagement and commitment (cognitive participation), facilitating the use of the intervention (collective action) and the value of the intervention (reflexive monitoring).

Interview and focus group discussions were audio recorded and transcribed verbatim by an external organisation. The audio recordings and transcriptions were not shared with anyone outside the research team. The transcribed data were uploaded into NVivo for data management and analysis.

## Observations
During the implementation phase, AC and RP also conducted ethnographic-type observations in the homes to understand current practice and to explore possible changes due to the intervention; this included collecting data through informal conversations with staff and observations of activities related to the implementation of the intervention. Brief field notes were made during each visit and written up immediately afterwards and uploaded and managed in NVivo.

## Analysis
### Quantitative data analysis
We summarised quantitative data as descriptive statistics in tables and charts as appropriate.

### Qualitative data analysis
Qualitative data analysis was based on the constant comparison method.[16] Codes created from themes generated from the data were deductively mapped to a priori concepts and components of normalisation process theory. Researcher bias was minimised through regular cross-checking of data and findings by the members of the research team. Anonymised quotes have been used as exemplars of key points.

## RESULTS
### Quantitative data
Quantitative data are presented in relation to the four key components: context, dose delivered, reach and dose received.

### Context
Data were collected prior to implementation of the intervention. There were four nursing homes and two residential homes (without nursing care); the number of beds ranged from 32 to 62, with bed occupancy between 58% and 100%. Over three-quarters of residents were female. The age range for male residents was 63–96 years, and 57–103 years for females. There were 265 staff, consisting of managers, nurses, senior and junior carers and ancillary staff (table 2).

### Dose delivered
We had aimed to provide four training sessions in each home, two for senior care staff (nurses and senior carers in nursing homes and senior carers in residential homes) and two for junior care staff. We ran 21 training sessions across the six homes, lasting a total of 35 hours, (range 4–9.5 hours per home). In some cases, it was necessary to combine junior and senior training sessions into a single event as insufficient staff were available to attend their designated session. Follow-up training for new staff or those unable to attend the initial training was delivered by REACH Champions or a member of the study team using the training video and study handbook.

### Reach
The number and grade of staff who attended the initial and follow-up training sessions are shown in table 3. We trained 101/219 (46%) of all care staff (we did not provide training for ancillary staff) in the homes, 35/42 (83%) of nurses, 17/25 (68%) of senior care staff and 45/143 (31%) of junior care staff.

### Dose received
The 'use of decision-making algorithm' form was completed 135 times across five homes during the 6-month implementation phase of the study; one home was unable

**Table 2** Characteristics of the participating care home and residents prior to intervention implementation

| | Northern Ireland | | | England | | | Total |
|---|---|---|---|---|---|---|---|
| Home | A | B | C | D | E | F | |
| Type of home | Nursing | Nursing | Residential | Nursing | Nursing | Residential | – |
| No of beds | 62 | 32 | 36 | 56 | 51 | 40 | 277 |
| Bed occupancy (%) at baseline | 36 (58) | 26 (81) | 36 (100) | 42 (75) | 51 (100) | 36 (90) | 227 (83) |
| Male residents (%) | 12 (33) | 8 (36) | 5 (14) | 12 (29) | 13 (25) | 5 (16) | 55 (24) |
| Female residents (%) | 24 (67) | 18 (64) | 31 (86) | 30 (71) | 38 (75) | 31 (84) | 172 (76) |
| Age males (years, median) | 75.5 | 83 | 80 | 82 | 86 | 82 | – |
| Age females (years, median) | 85.5 | 88 | 87 | 92 | 89.5 | 85 | – |
| No of staff (includes ancillary staff) | 67 | 39 | 18 | 53 | 38 | 50 | 265 |

to complete any forms primarily due to staffing shortages. Staff reported using the decision-making algorithm 81 times. UTIs were the most common suspected infection in residents (49%), with RTIs ranked second (28%). The most frequently reported reasons for not using the algorithm were that other tests were carried out or the resident was too unwell.

## Qualitative data
### Focus groups and interviews
There were 12 preimplementation focus groups, two in each home (one for staff and one for relatives of residents), involving a total of 41 staff and 28 relatives. We conducted semistructured one-to-one interviews with eight GPs (five in NI and three in the West Midlands).

There were six postimplementation focus groups, with one focus group conducted in each care home, involving 26 staff in total. We conducted semistructured face-to-face interviews with six REACH Champions and five managers of homes (one manager had left the home at the time of the interviews).

The findings from the focus groups, interviews and observational field notes are presented based on the four main components of normalisation process theory.

### Making sense (coherence)
Here, we report how participants understood the problem the intervention aimed to address, how they perceived their use of the intervention to impact this problem, and how they understood what they were being asked to do differently from their usual practice.

Preimplementation care home staff and relatives reported varied views regarding how they understood the problem of AMR and how the intervention could address it. Participants described how they thought antibiotics were often too easy to access either from the GP or from on-line sources, and were thus contributing to AMR. Staff described how relatives sometimes insisted on antibiotics being prescribed if they thought their relative had an infection.

Postimplementation, staff reported how the training led them to reflect on their own personal use of antibiotics and improved their knowledge of the local and global AMR problem.

> What I liked about [the training] was knowing that it is a global problem, that the effects of overprescribing of antibiotics is affecting us without us really realising because we tend to ask the GP for antibiotics for everything that we think that is happening to the resident, (…) which will be difficult later on in life because then you won't have anything to fight the bacteria. (Champion, Post-Implementation)

Some staff described how using the decision-making algorithm made them focus more on preventative measures (eg, encouraging fluid intake) before contacting a GP for a suspected infection. Although staff recognised that the algorithm may impact prescribing in their own home, it would not have a wider impact without a change in attitude in the general public. It was also reported that GPs would continue to prescribe antibiotics for sick

**Table 3** Number and grade of staff attending initial and follow-up formal training

| Type and no of staff (excluding ancillary staff) | Initial training no (night staff) | Follow-up training (night staff) | Total no trained (night staff) | % of care staff trained |
|---|---|---|---|---|
| Nurses n=42 | 29 (10) | 6 (2) | 35 (12) | 83 |
| Senior care staff n=25 | 16 (4) | 1 (0) | 17 (4) | 68 |
| Junior care staff n=143 | 40 (8) | 5 (0) | 45 (8) | 31 |
| Managers n=9 | 2 (0) | 2 (0) | 4 (0) | 44 |
| **Total n=219** | **87 (22)** | **14 (2)** | **101 (24)** | **46** |

residents who had a past history of infection in order to avoid risk of rapid deterioration.

## Engagement and commitment (cognitive participation)

Managers reported different reasons for taking part in the study, which included, contributing to a potential solution to the problem of AMR and to give staff an opportunity to gain knowledge to improve the quality of care to residents.

I always feel anything to empower us, to give us more knowledge and to participate, to improve quality is worth it for us. (Manager, Post-Implementation)

In most homes the manager appointed the REACH Champion with their agreement, who was either a nurse or senior carer. However, some managers reported challenges in appointing a champion, for example, in one small nursing home with many part-time staff, the manager reported difficulties in appointing someone to this role.

The problem is, I have link nurses for lots of different topics and I have run out of people to give responsibility to and [NAME] didn't actually want to take it on either, but I mean somebody had to do it. (Manager Post-Implementation)

Postimplementation, no champion reported being unhappy in their role and some reported that they perceived their role as one which enabled them to be part of a solution to an important problem, increasing their awareness about infection in the home and being something different to their routine work. The champions and staff described a range of tactics used to engage staff. This included challenging staff at opportune moments to consider if it was necessary to contact the GP immediately; and discussing the algorithm and documentation in formal situations such as handover, regular staff meetings and in casual break-time conversations.

[REACH Champion] helped us with the meetings with you, or to ask us '[NAME], what you've done, do you think of things like that?' Or just to remind us all the time when we are sitting and talking. (Staff, Post-Implementation)

However, the pressures of everyday work in some homes meant that engagement could be challenging at times, for example, completing study documentation. Frequent and regular visits by the researchers were seen to be encouraging and helpful to the champions and staff.

The REACH training was reported as useful in providing insight into the problem of AMR and highlighting what was expected from staff throughout the study.

In the training, we got an idea because you give the examples, the case studies, and we discussed it and we found out how we are going to do it through REACH. (Staff, Post-Implementation)

In residential homes, staff were provided with additional training to take temperatures using thermometers, as this is not part of their usual practice. They described how the training they received to do this was very helpful in communicating information about a suspected infection to a GP.

[The Champion] …the GP always ask if the resident has a temperature and now they are able to say what it is instead of just saying that the resident is warm or clammy. (Observation, Month 6)

Staff in residential homes also suggested that it would be useful to have training on how to deal with relatives' concerns, especially for residents prone to infection.

Supporting training materials such as the DVD and handbook (part of the training component) were used in some cases, but staff reported their preference for face-to-face training.

The study handbook was given to each staff member to have it for consultation at home, and I do believe it will be well-used afterwards. Any time that someone has a question or a doubt, they can go and check the book. (Champion, Post-Implementation)

## Facilitating the use of the REACH intervention (collective action)

Staff described how their prior knowledge of the resident impacted on whether they used the algorithm. For example, staff would contact the GP immediately for those residents with a history of septicaemia.

With [resident], we had a history of going into septicaemia very quickly. You don't take a risk like waiting for the symptoms to come again (…) we didn't get the chance to look at the aid. (Staff, Post-Implementation)

The decision-making algorithm was reported to be of more value in situations where staff lacked this prior knowledge of the resident; for example, when there was either a new member of staff or a new resident.

### Knowledge and experience around UTIs

Staff described how the algorithm influenced their assessment and management of UTIs and their decision to contact the GP. Some reported how they were now less likely to rely on foul-smelling or strong coloured urine or a urinalysis as indicators of infection and more alert to the symptoms indicated on the algorithm.

We are lessening that time we are phoning the GP because we are not prompted just with that urine dipstick. Unlike before, that is the first thing that I usually just check and then it is like that is your cue to talk to the doctor and now you just think maybe I have to keep an eye and then just check for any other symptoms. [Staff, Post-Implementation]

However, staff reported that they found the decision-making algorithm most challenging to use for UTIs, because the urinary symptoms stated in the algorithm (new or increased frequency, urgency or incontinence, blood in urine and lower abdominal pain) were not applicable for many residents in care homes, particularly those with dementia who were also incontinent. Therefore, staff were more likely to ignore the decision-making algorithm for these residents as many of the UTI symptoms were deemed irrelevant. Staff also described how they needed to be more vigilant for these residents, usually telephoning the GP to avert the risk of residents deteriorating quickly.

### Knowledge and experience around RTIs

Most staff thought the algorithm worked well for RTIs. This was because it reflected the symptoms they usually looked for and aligned with other aspects of their practice. Nurses reported finding RTIs easier to manage than UTIs because they perceived there to be more actions to take, including using nebulisers, encouraging fluids and monitoring observations.

> Like I said to you on the chest infection, it was necessary only to look for one or two and it was there, and it really worked. Like it was really, really good the way it was written in here. So, we could really rely on it. [Staff, Post-implementation]

### Knowledge and experience around SSTIs

In general, staff reported that they used the decision-making algorithm less for SSTIs than for other infections. This was because SSTIs were less common, the symptoms were very easy to observe, and they were able to act quickly to prevent them from getting worse.

> With the skin symptoms I don't think we did [use the algorithm]. Usually when we have a wound they don't really get worse. We tend to catch them very early and then we try to prevent them from getting worse. [Staff, Post-Implementation]

### Interaction of the intervention with colleagues and GPs

Some staff reported ways in which using the algorithm strengthened their communication with the GP when they suspected a resident had an infection. The algorithm appeared to work like a checklist helping staff to justify when to contact a GP if they perceived the resident to have symptoms according to the algorithm, or delay contacting the GP if the resident was not presenting with symptoms. The algorithm also served as a tool to persuade GPs to visit the resident and to give staff more 'vocabulary' of what to say and a structure for saying it.

Some junior staff felt empowered in being able to go to a senior member of staff with clearer information on a resident who was unwell, while others felt it was not the role of junior staff to do this, and it was not within their role to contact GPs.

### Interaction of the intervention in communication with residents or relatives

Staff described how they were required to comply with a resident's or relative's demand for action, such as contacting the GP when the resident themselves suspected they may have had an infection.

> The families don't know about decision aid tool but sometimes, we can get pressured for antibiotics. If they understand the person is unwell, 'why are they not on antibiotic?' [We say] 'Sometimes it's not needed or sometimes it will harm more than benefit'. … Yes, families want the treatment. No treatment may appear as no action that 'nothing is being done', and nothing as being done is, emptiness, its lack of action, its neglect almost. [Champion, Post-Implementation]

### Burden of data collection

Staff reported that completion of a 'use of decision-making algorithm' form each time they suspected a resident may have an infection time consuming and burdensome and they did not always remember to complete the forms. Staff understood the importance of completing the form for the study, but considered that it duplicated information already routinely recorded by the home and therefore added to their workload and burden of documentation.

### Value of the intervention (reflexive monitoring)

There were differing opinions regarding which staff, with which skill set and responsibility, should use the algorithm. Nurses and senior carers in residential homes, who had responsibility for contacting the GP, discussed how the algorithm acted as a reference for symptoms of infection and actions to take. They perceived it to be particularly useful whenever there were elements of unfamiliarity, for example, new or inexperienced senior staff, or when dealing with unfamiliar GPs. However, some senior and junior care staff in nursing homes reported that the algorithm was of no value to them. They emphasised that because their role was limited to reporting concerns to the nurse, a tool to help nurses know when to contact the GP was irrelevant to them and beyond their usual responsibility. Additionally, one champion in a nursing home described how senior and junior care staff lacked both initiative and competence to be able to use the decision-making algorithm.

## DISCUSSION

Our objective in this process evaluation was to describe the facilitators and obstacles to the implementation of the REACH intervention.

We delivered a training package to a substantial number of the key staff in the homes, which was generally well received. Training in care homes is challenging but can bring about changes to practice.[17] A larger study of longer duration may require more frequent training to ensure continued engagement with the use of the

decision-making algorithm. It can be difficult for care homes to release staff to attend training sessions, and a flexible approach to delivering training is important to optimise attendance. We did provide a DVD format which staff could have watched in their own time which provided some flexibility. Integrating the algorithm (tool) into technology (such as an iPad) may have promoted and streamlined decision making, but we did not have the resources to provide this approach. However, it could be considered in the future.

Using normalisation process theory to frame the analysis gave us an interesting insight into implementation of the decision-making algorithm into practice over a 6-month period.

Making sense (coherence), preimplementation discussions reflected findings from a systematic review of the public's knowledge and beliefs about AMR with some participants reporting limited understanding of the problem, while others were more knowledgeable.[18] This suggests the need for more work in care homes to raise awareness. Postimplementation results revealed staff to be generally more knowledgeable about AMR but they were unsure if the intervention would change how GPs prescribe antibiotics.

We did not see a lack of coherence which is often cited as an important challenge in the implementation (ie, the intervention does not make sense or is met with conflicting attitudes.[19 20]

Engagement and commitment (cognitive participation) were generally high, with managers and carers feeling more empowered.[21] Evidence from multiple studies shows that a sense of empowerment can facilitate the implementation of beneficial practices.[22–24] Empowerment contributes to a sense of worth and generates greater aspirations.[25 26]

Champions in research settings are seen as invaluable.[27] In this study, although some managers described challenges in appointing a Champion, generally our analysis suggested that they were very important in helping to ensure the engagement and commitment of care home staff. Similarly, although champions spoke positively about the role, they also described challenges related to time commitment and burden of study paperwork which may have hindered their effectiveness. Thus, implementing the REACH algorithm in care homes may require particular consideration of how Champions may be resourced to conduct their role.

In Facilitating the use of the REACH intervention (collective action), our results were somewhat inconsistent. There was evidence that some staff were using the decision-making algorithm but others were not. In some cases, staff forgot to refer to it, and in others, staff reported not having time. In the case of the latter, this was partly related to staff having to complete additional forms if they used the decision-making algorithm. It seems that the REACH intervention provided access to learning for the teams within the homes, but implementation was mediated by the context of working in a care

home environment as seen in other studies.[28] Finding the time to complete study-specific paperwork is a challenge encountered in many studies.[29]

Staff reported that some of the symptoms included in the decision-making algorithm, despite being evidence based, were not easy to detect in residents with dementia or with urinary incontinence and therefore chose to use their usual methods of assessment. Some staff did not use the decision-making algorithm, noting that their own knowledge of the resident (through close contact from delivering care) was more important.

Staff described how a relative's request to contact the GP could overrule their own decision of when they consider this to be necessary. Medical situations involving decisions with relatives about ill residents are very challenging, as we have seen in this study. Trust and confidence can be seriously challenged and conflicts may arise.[30 31] Van Keer *et al* suggest that families may have unrealistic expectations of care, perhaps based on media misinformation and lack understanding of a prognosis.[32] Raising awareness and providing clear information about prescribing decisions may help, and training on how to handle discussions with relatives may also be useful. Engaging with family members on the topic of AMR and the use of the decision-making algorithm may improve knowledge and acceptance, either through face-to-face meetings, newsletters or notices in the care homes.

Value of the intervention (reflexive monitoring) reflected a more negative outcome than the other constructs. While most supported the approach, operationalising it was more problematic. Some reported that junior staff lacked the skills and competence to use the algorithm and some nursing staff considered that it was beyond the accepted role of some levels of staff. However, we found evidence that some junior care staff found the decision-making algorithm useful to decide when to alert a senior member of staff to change in a resident's condition.

Reflexive monitoring highlighted a number of challenges in REACH; while valued by staff, it was not fully implemented. In a feasibility study, this may be expected as the implementation period was short, while in other studies, such periods have been longer, allowing for a new practice to be normalised.[18] It has been argued that for practices to become accepted, integrated, and sustained in day-to-day work, they must be experienced in dealing effectively with real everyday problems.[33]

Since we completed this work, new research has indicated that older people presenting to their GP with a confirmed, or suspected, UTI who are not prescribed antibiotics, are more likely to die.[34] The work does not distinguish between care home and community-dwelling older people. It has important implications for how suspected UTIs are defined. Record studies of this nature do not allow the level of suspicion to be measured. Our algorithm identifies those where there is a reasonable level of suspicion for referral to GP. This does not equate directly to a diagnosis of suspected UTI. Nevertheless, it

is possible that an approach seeking to reduce care home referrals to GPs for suspected UTI might increase overall mortality in care home residents with suspected UTI, while reducing overall AMR and reducing mortality from infectious diseases over a longer period in a wider population. More data are needed to help society balance these competing risks and benefits.

The process evaluation used a mixed-methods approach which generated a rich and wide-ranging understanding of the facilitators and obstacles to implementation of a new intervention into practice. However, it is important to note the study took place in six care homes in the UK, therefore, all findings must be interpreted in the light of this small sample. Similarly, the caveats associated with qualitative work must be considered that is, findings may not be generalisable.

In conclusion, we were able to deliver a training package to a substantial number of key staff in care homes. A decision-making algorithm for common infections in care homes empowered staff but could be challenging to operationalise at times and staff questioned whether if it would influence prescribing for infections or impact AMR. Despite being evidence-based, staff felt that some of the symptoms included in the decision-making algorithm, were not relevant for residents with dementia or with urinary incontinence. Any future study should consider the findings from the process evaluation to help to ensure the success of implementation on a larger scale.

**Author affiliations**
[1]Clinical Trials Unit, Warwick Medical School, University of Warwick, Coventry, UK
[2]Health Protection Research Unit, Imperial College London, London, UK
[3]University Hospital Coventry and Warwickshire, Coventry, UK
[4]School of Pharmacy, Queen's University Belfast, Belfast, UK
[5]Northern Ireland Clinical Trials Unit, Royal Hospitals, Belfast, UK
[6]Centre for Public Health, Queen's University Belfast, Belfast, UK
[7]Department of Pathology and Molecular Medicine, McMaster University, Hamilton, Ontario, Canada
[8]Orchard Care Homes, Harrogate, UK

**Acknowledgements** We would like to acknowledge and thank the contribution of the patient advisory group and all study participants including relatives, care home staff and GPs. In particular, we are grateful to the REACH Champions for their support throughout the study.

**Contributors** RP: drafted the manuscript, contributed to the study design, acquisition, analysis and interpretation of data. AC: contributed to the study design, acquisition, analysis and interpretation of data and critical revision of the manuscript. DRE: developed the initial concept of the study, drafted the manuscript, contributed to the study design and interpretation of data. CS: contributed to the study design, interpretation of the data and critical revision of the manuscript. EG: contributed to the analysis of the data, interpretation of the data and critical revision of the manuscript. AA: contributed to the analysis of the data, interpretation of the data and critical revision of the manuscript. DO: contributed to the study design, interpretation of the data and critical revision of the manuscript. MU: developed the initial concept of the study, contributed to the study design, interpretation of the data and critical revision of the manuscript. ML: contributed to the study design, interpretation of the data and critical revision of the manuscript. BS: contributed to the study design, interpretation of the data and critical revision of the manuscript. MT: contributed to the study design, interpretation of the data and critical revision of the manuscript. CH developed the initial concept of the study, interpretation of the data and critical revision of the manuscript.

**Funding** This work was funded by the National Institute for Health Research (NIHR), grant number HS&DR 13/97/12.

**Competing interests** CH is a member of the Health Services and Delivery Research Commissioned Panel. ML has worked for the WHO as a consultant to develop antibiotics for an essential list of medicines and algorithms for appropriate antibiotic use. MU is a member of National Institute for Health Research (NIHR) Journals Library Editors Group. Chairperson of the National Institute for Health and Care Excellence accreditation advisory committee from 2013 until March 2017, for which he received a fee. Chief investigator or coinvestigator on multiple previous and current research grants from NIHR and Arthritis Research UK and is a coinvestigator on grants funded by Arthritis Australia, Australian National Health and the Medical Research Council. He has received travel expenses for speaking at conferences from the professional organisations hosting the conferences. Director and shareholder of Clinvivo Ltd (Tenterden, UK), which provides electronic data collection for health services research. He is part of an academic partnership with Serco (Hook, UK) related to return-to-work initiatives. Editor of the NIHR journal series, for which he receives a fee. He has accepted an honorarium for advice on Research Excellence Framework submission from Queen Mary University of London. He is coinvestigator on an Efficacy and Mechanism Evaluation grant, receiving support in kind from Orthospace (Caesarea, Israel).

**Patient consent for publication** Not required.

**Ethics approval** The study was granted ethical approval by the Office of Research Ethics Committees Northern Ireland Health and Social Care Research Ethics Committee B (Ref: 16/NI/0003).

**Provenance and peer review** Not commissioned; externally peer reviewed.

**Data availability statement** Data are available on reasonable request.

**ORCID iDs**
David R Ellard http://orcid.org/0000-0002-2992-048X
Ashley Agus http://orcid.org/0000-0001-9839-6282

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
