## [Reviewer comments · BMJ Open]

ARTICLE DETAILS

TITLE (PROVISIONAL)	A multifaceted intervention to reduce antimicrobial prescribing in care homes: a process evaluation of a UK based non-randomised feasibility study
AUTHORS	Potter, Rachel; Campbell, Anne; Ellard, David; Shaw, Catherine; Gardner, Evie; Agus, Ashley; O'Reilly, Dermot; Underwood, Martin; Loeb, Mark; Stafford, Bob; Tunney, Michael; Hughes, Carmel

VERSION 1 – REVIEW

REVIEWER	Martin Cormican NUI Galway Ireland
REVIEW RETURNED	06-Jul-2019

GENERAL COMMENTS	This is an interesting paper addressing the very practical problem of factors that influence implementation of approaches to influence antimicrobial prescribing in care home settings. It will be of interest to people developing projects to address this. The need to go beyond awareness and training to address workplace context issues relating to time and staffing is important. The principle concern i have about this paper is i do not see that the issue of research ethics is adequately addressed - how were care home workers recruited, did they consent and what measures were taken to ensure that their views were not influenced by or visible to their employers. I note the the paper indicates that in most homes the REACH Champion was appointed with their agreement - but goes on to refer to issues regarding difficulty "persuading staff to take on responsibility". It would be helpful to address in a revision how what protections were in place to ensure that the persuasion was reasonable and if people who participated in the process were similarly persuaded
---

REVIEWER	Morgan Katz Johns Hopkins University Baltimore MD, USA
REVIEW RETURNED	08-Jul-2019

GENERAL COMMENTS	Excellent design and description of the very real problems with which many facilities are faced when attempting to improve antibiotic stewardship practices. Consider adding a bit more discussion regarding interventions that may improve the feasibility of the training process or increase uptake for the staff. For example, any thoughts on reducing burden of the tools and making part of the workflow, or engaging residents and family members who may demand antibiotics?
--

VERSION 1 – AUTHOR RESPONSE

Reviewer 1

In response to the reviewer 1 comments we have added the following to the manuscript:

Ethical approval was granted by the Office of Research Ethics Committees Northern Ireland Health and Social Care Research Ethics Committee B (Ref: 16/NI/0003).

Care home managers were asked to distribute information packs on behalf of the research team to care home staff which included a participant information sheet, consent form and letter inviting them to take part in the study. Participation was voluntary and written consent was obtained from all participants. The managers of the care homes were not invited to take part in the focus groups so as to prevent the management relationship directing or constraining the group discussion.

All interviews were transcribed and anonymized and all quotations given anonymized labels.

We have added the following to the manuscript to describe the role of REACH Champion and to make it clearer to the reader that any persuasion by the home managers to appoint a REACH Champion was reasonable. We also added a short section in the discussion to explore the role of Champion in future studies.

On agreeing to take part in the study, the manager of each care home was asked to appoint a REACH Champion, an individual who would be responsible for delivering training to staff unable to attend the original REACH training session

In most homes the manager appointed the REACH Champion with their agreement, who was either a nurse or senior carer. However, some managers reported challenges in appointing a Champion e.g. in one small nursing home with many part-time staff, the manager reported difficulties in appointing someone to this role.

The problem is, I have link nurses for lots of different topics and I have run out of people to give responsibility to and [NAME] didn't actually want to take it on either, but I mean somebody had to do it. (NI.NH.02: Manager Post-Implementation)

Post-implementation, no Champion reported being unhappy in their role and some reported that they perceived their role as one which enabled them to be part of a solution to an important problem, increasing their awareness about infection in the home, and being something different to their routine work.

Champions in research settings are seen as invaluable. In this study, although some managers described challenges in appointing a Champion, generally our analysis suggested that they were very important in helping to ensure the engagement and commitment of care home staff. Similarly, although Champions spoke positively about the role, they also described challenges related to time commitment and burden of study paperwork which may have hindered their effectiveness. Thus, implementing the REACH algorithm in care homes may require particular consideration of how Champions may be resourced to conduct their role.

Reviewer 2

In response to Reviewer 2 comments we have also added the following to the discussion:

It can be difficult for care homes to release staff to attend training sessions, and a flexible approach to delivering training is important to optimise attendance. We did provide a DVD format which staff could have watched in their own time which provided some flexibility. Integrating the algorithm (tool) into technology (such as an iPad) may have promoted and streamlined decision making, but we did not have the resources to provide this approach. However, it could be considered in the future.

Engaging with family members on the topic of AMR and the use of the decision-making algorithm may improve knowledge and acceptance, either through face-to-face meetings, newsletters or notices in the care homes.